# What Are ILK in Relation to Science? Using the 'Ethic of Equivocation' to Co-Produce New Knowledge for Conservation

Meredith Root-Bernstein [1,2,3,*], Pierre du Plessis [4], Matías Guerrero-Gatica [2,3,5], Trupthi Narayan [6], Samuel Roturier [7] and Helen C. Wheeler [8]

1. Musée National d'Histoire Naturelle, CNRS, 75005 Paris, France
2. Center of Applied Ecology and Sustainability, Santiago 8331150, Chile
3. Institute of Ecology and Biodiversity, Santiago 7800003, Chile
4. Oslo School of Environmental Humanities, University of Oslo, 0315 Oslo, Norway
5. Departamento de Geografía, Universidad de Chile, Santiago 8380000, Chile
6. Department of Geography, University of Paris Nanterre, 92000 Nanterre, France
7. Université Paris-Saclay, CNRS, AgroParisTech, Ecologie Systématique Evolution, F-91190 Gif-sur-Yvette, France
8. School of Life Sciences, Anglia Ruskin University, Cambridge CB1 1PT, UK
* Correspondence: meredith.root-bernstein@mnhn.fr; Tel.: +33-0140793034

**Abstract:** Attention to epistemological relationships between Indigenous and local knowledges (ILK) and conservation science is increasing. Some approaches for doing so have been developed, but in general, serious engagement between ILK and science still feels experimental and does not have broad uptake. Here we address some of the main issues that arise when considering the epistemological relationships between ILK and conservation or ecological science. A key issue is the relationship between ILK and science that sets the expectations for how they may form dialogues, collaborations, or co-production. After reviewing several perspectives on this relationship, and their limitations, we suggest that a focus on shared meanings within an 'ethic of equivocation' is a productive path forward. In an ethic of equivocation, neither ILK nor science validates the other, yet meaning can be created. We explain and develop the concepts of *meanings* and *equivocation* in the ILK-science context. We, thus, argue for a broad and rich understanding of the joint roles of ILK and science, which goes beyond treating ILK as a data source. We argue that diverse styles of reasoning exist in science, and scientists may already treat observations, models, and collaborations in the ways that we discuss, providing ready-made analogies for thinking about ILK. We also discuss how to avoid abuses of power while engaging with ILK to co-produce new knowledge.

**Keywords:** indigenous and local knowledges; conservation; ecology; equivocation; dialogue; integration; co-production

## 1. Introduction

The recent Intergovernmental Science-Policy Platform on Biodiversity and Ecosystem Services (IPBES) reports on the state of biodiversity are part of an important call for decision-makers to stop the deterioration of ecosystems. They also mark a milestone in the use of Indigenous and local knowledges (ILK) in high-level international science policy documents, the result of almost 30 years of research triggered by Article 8j of the Convention on Biological Diversity (CBD). IPBES set up an ILK task force with a clearly defined mandate, which was implemented by the Technical Support Unit (TSU) hosted at UNESCO. The TSU facilitated a series of dialogue workshops involving ILK holders and academics, and put together an "ILK library of materials as a resource for IPBES authors" [1,2]. This effort highlights the increasing prominence of science and ILK working together and the rapidly evolving discussion on the synthesis and use of science and ILK in decision-making.

The epistemological relationships between ILK, science, management, and policy currently have many forms. We use the term "**epistemological** relationship" (see Box 1 for definitions of terms in bold) as a broad formulation that points to how different forms of knowledge are formed and used ("epistemology") without specifying a model for their interaction ("relationship"). This is because some existing terms, such as "integration", have come to be associated with methods that imply an unbalanced relationship of power in which ILK may be subsumed by the scientific framework, detached from the producers of that knowledge. Not only can this be derogatory, but it also risks further marginalizing and disempowering ILK holders. The methodological pluralism resulting from the range of epistemological relationships reflects the many different socioecological contexts in which ILK meets Western conceptions of conservation and management, and the many questions ILK may address. Undoubtedly, there is not one right way to engage with ILK, due to the diversity of these knowledge systems and their multiple relationships to varying social, political, and ecological contexts, and different scientific endeavors [3]. At the same time, different methods and philosophies for engaging ILK create different kinds of epistemological relationships, different forms of value, and different ways in which knowledge can be used in the world.

**Box 1.** Definitions of some key terms.

---

**Cosmology**: a culture's claims about the kinds of things in the world (e.g., species, materials, spirits, gods, etc.), their natures or properties and capabilities, and their distribution.
**Epistemology**: A system of reasoning about what constitutes knowledge, the desirable properties of knowledge, and how knowledge is formed. This is usually associated with a particular cosmology and ontology. For example, the epistemology of science can be characterized by claims that knowledge is formed through the experimental method, and should strive to be universal, objective, reductionist, falsifiable, etc.
**Equivocation**: a partial, contextual alignment between two concepts or practices that are not the same, and which actors may be aware are not the same, but which form a functional basis for interaction and exchange.
**Holistic–mechanistic distinction**: Holistic thinking attempts to understand various aspects of a complex whole, simultaneously. Mechanistic thinking attempts to break the whole into parts and to understand how each part behaves in a causal process.
**Ontology**: a description of a culture's understanding of the origins and structure of the world, and the kinds of interconversions and relationships between the things defined by cosmology. In short, "how the world works." For example, a simplified scientific ontology claims matter originated with the Big Bang and subsequently developed into two kinds, abiotic and biotic, which are recycled back and forth through biogeochemical processes.

---

We direct this paper towards ecologists and conservation biologists (students, academics, and professionals) in an effort to encourage more meaningful, in-depth approaches for engaging with ILK in their research. We are a diverse group of non-Indigenous researchers speaking to the scientific research community. We should not be understood to represent, or to speak in place of, Indigenous scientists and non-scientist knowledge holders. We speak to scientists not because the engagement with ILK by scientists is a privileged or standard form of epistemological relationship, but because ecologists and conservationists are increasingly encouraged to engage with ILK, yet for many, this may still feel experimental and beyond a science skill set (e.g., [4]). Discussions of epistemological relationships with ILK are common within the literature on socioecological, biocultural, ethnobiological, or environmental anthropology interdisciplinary approaches (e.g., [5–8]). While interdisciplinary interaction with anthropologists is valuable, our goal in this paper is to suggest that ecological and conservation science can engage directly with ILK and that conservationists who are willing to be self-reflective about their methods and their thinking processes can develop rich epistemological relationships with ILK holders. These relationships should go beyond a vision where ILK serves science, to see science as also serving ILK, and together serving better knowledge of and action in the environment [9].

We start this paper with a short review aimed at ecologists and conservation biologists who do not have a background in anthropology and may not feel very confident about what ILK consist of. We address the simple but difficult question, "what are ILK?" By this, we do not mean a list of all the things that may qualify as ILK, but rather a general description. We

attempt to bring some clarity to the set of debates over what a general description of ILK would be. The question must be resolved in some way in order to ask the corollary, "How are ILK and science related, or how can they be brought into relation?" If we were going to express this in a simplistic and reductive way, we would ask whether ILK and science are the same or different—but as we hope to illustrate, this question cannot be satisfactorily answered. The question of the relation between science and ILK may sound philosophical, and perhaps irrelevant to the action-oriented "crisis discipline" of conservation. However, the relationship between ILK and science sets the expectations for how they may form dialogues and collaborations, co-productions, or other methods that put epistemological relationships into action (see e.g., [10–13]). Throughout the paper, we use analogies and metaphors to try to help ecologists and conservation biologists think about ILK in ways they may not have before. After reviewing several perspectives on what ILK are and how they are similar or different to science, we suggest that the limitations of most existing answers to these questions can be overcome with the 'ethic of equivocation', an approach that comes out of South American anthropological theory, as we describe below. While there may be a perspective that the ethic of equivocation may be specifically South American in its concerns and framing of the issues, similar ideas have been developed in other contexts, and we believe it can be successfully adapted outside of South America.

## 2. How Are ILK Different from Science, and So What?

What is knowledge? Although there is no standard definition, even in Western cultures many people would agree that knowledge takes many forms, including as art, as manual skills, or as feelings. Nevertheless, in academic and formal Western settings, knowledge is valued when it is transmitted in particular formats of words and numbers, such as scientific papers. This affects how scientists often think about ILK. Knowledge can also be thought of as necessarily including a continuous *process* of learning and engaging with the unknown, or simply as discrete *outcomes* of that process, such as facts, for example, should you separate scientific facts from the scientific method when teaching science? As we will see, this distinction between knowledge processes and outcomes is not always clarified and can lead to misunderstandings.

The varieties of ILK are widely believed to be different from science (see [14]). In the past, ILK has been characterized as non-analytic, dogmatic, complacent about contradictions, non-innovative, and closed to other worldviews—a view that was subsequently critiqued as not inherently or generally the case [15]. Rather, any form of knowledge can become dogmatic, complacent, and closed, and the history of science is full of such examples (e.g., [16]). Similarly, in early literature, ILK were often characterized as an artifact—a sacred body of knowledge completed in the ancient past [17]. This view, solidified by anthropologists in the mid-20th century, has since been thoroughly critiqued and discredited by more recent anthropological research (e.g., [18,19]). Today, ILK are much more likely to be characterized as dynamic, adaptive, and open to incorporating modern technologies and science ideas [5,8,20,21].

This shift toward de-emphasizing an opposition between ILK and Western science in favor of emphasizing ILK's dynamic and adaptive qualities is not total, however. For example, Whyte et al. [22] envision ILK as relational, contextual, anecdotal, and embedded in a worldview of morality and animism, which they contrast with a universal Western scientific knowledge system predicated on emotional distance and exploitation of an inanimate, mechanistic world. However, in practice, neither scientists nor science are unemotional, uninfluenced, uninterested, or detached (e.g., [23,24]). Both ILK and science are dynamic, engaged, and adaptive because both have social and intellectual histories and are embedded in diverse societies (e.g., [25]).

Several other objections to a dualist view of ILK vs. science have also been advanced in the literature. Ludwig and Poliseli [26] argue that this kind of contrast rests on a **holistic–mechanistic distinction**, which they argue is a false distinction. Contrary to this holistic–mechanistic distinction, they argue that scientists can be holistic thinkers too, and

ILK can embed mechanistic causal analyses inherently in the form of management practices, values, and worldviews (e.g., [27]). However, it is because causality may be embedded in cultural forms that, according to Ludwig and Poliseli [26], ILK lose meaning when extracted from their cultural contexts (see also [20,28]). An analogous loss of context in science would be extracting the data points from papers in order to do a meta-analysis, without reading the methods sections to see if the studies are really comparable to one another. While making this argument, Ludwig and Poliseli [26] accept that science is universal and general. Green [6] (see also [15]) explicitly challenges this universality vs. contextuality as a valid distinction between science and ILK, arguing that models and data are also contextual, being true only in particular scales or frames. Quantum physics does not apply to stoichiometry; stoichiometry does not explain convergent evolution; etc.

In light of these debates, in which a series of proposed ILK-science distinctions are sequentially cast in doubt, perhaps it is not surprising that some researchers take the view that any differences within ILK and between ILK and science have limited relevance, and universal commonalities should be emphasized (e.g., [29]; see [30]). Löffmarck and Liedskog [28] argue that the IPBES framework also takes this perspective with its claims that all knowledge systems represent a common universal reality and that ILK is just science that is complicated to validate (see [31,32]). The IPBES resolve this complicatedness by claiming that each knowledge system is self-validating, creating equally valid strands of evidence that can be synthesized [33,34]. However, there are challenges related to this act of synthesis that have not yet been fully resolved [28]. Synthesis of knowledge from multiple worldviews will not, by definition, be intellectually neutral. Synthesis of evidence is an analytical, epistemological process: it requires knowledge-as-process. Knowledge cannot exist without a worldview. The worldview provides the implicit and explicit means to interpret and act on observations, methods, and analyses. Documented cross-cultural differences in how we understand the world [35] are the fundamental underpinnings of knowledge. It is increasingly understood that these differences in worldviews are meaningful and necessary for knowledge interpretation and action. We fail to take into account the interpretive role of the worldview if we imagine that a neutral power balance across multiple worldviews engenders a neutral form of knowledge with a basis in nobody's worldview. There is no position outside of all worldviews from which to make a neutral form of knowledge. We, and others [6,28], consider that any claim of universal commonality across ILK and science amounts to a claim that the distinctive worldviews of ILK are not important for interpreting their observations of the world, a claim that makes no sense. We suggest that many approaches to synthesis that use multiple strands of evidence from ILK and science are either implicitly synthesizing that knowledge under a single (science) worldview, or implicitly using the ethic of equivocation without perhaps having a name for it (see below).

To think about the issues in the preceding paragraph through an analogy, a claim of universal commonality across ILK and science is akin to saying that data on plant richness produced using 1 m square quadrats placed at fixed distances from termite nests can legitimately be reanalyzed *as if* it had been collected using remote sensing on systematically placed transects of 1 km × 50 m. Obviously, such a reanalysis makes no sense, and neither does the proposition that ILK can be understood *as if* it had no significant differences from science. Ecological data are only interpretable with reference to the sampling strategy and methods with which they were collected, just as ILK and science are only interpretable within the worldviews from which they emerged. In both cases, means of comparison do exist: to compare the results of both studies of termite nest effects on plants, one could do a meta-analysis. In this analogy, the ethic of equivocation, which we will introduce in a moment, has the same role as meta-analysis.

To reiterate, the critical differences and similarities between ILK and science are not found in qualities, such as contextuality, dynamism, morality, etc. Rather, the critical differences and similarities are found among **epistemologies, ontologies, and cosmologies**–the fundamental underpinnings of how people reason and construct understandings about

the world. This is analogous to how bilby species and rabbit species have phenotypic traits that are both similar and different, yet we do not understand their relationships according to convergence or divergence in ear and leg morphology alone (similar to how an 18th- or early 19th-century natural historian might have done [36]). What allows us to understand the relationship between bilbies and rabbits is the fundamental underpinnings of biogeography, phylogeny, and evo–devo (evolution of development). What allows us to understand the relationships between ILK and science are the fundamental underpinnings of epistemologies, ontologies, and cosmologies.

Further, the problem with looking for universal commonalities is the focus on comparing ILK *as a unitary whole* to science *as a unitary whole* [6]. By analogy, behavioral ecology, ethology, and cognitive science are all "about" animal behavior but are different fields. While some of their predictions can be made coherent, each of these subdisciplines has rich, non-overlapping research lines. A better approach to answering questions about the comparability of ILK and science may be to find particular *parts* of knowledge systems that coincide or complement one another [7,37]. A problem-led or question-led approach can avoid bias in the choice of which parts to focus on (see section "How to do equivocation in practice"). In summary, if there is no agreement that ILK and science are truly different, or that they have any fundamental commonality, we can come to understand that this is because 'are they the same or different' is fundamentally the *wrong question*.

### 3. The Ethic of Equivocation Described

The ethic of equivocation helps us work with ILK by identifying partial overlaps of knowledge systems, but without having to resolve issues of sameness and difference ([38–41], for related ideas see also [30,42]). Viveiros de Castro [38] argues for **equivocation** as the basis of all communication, translation, and cross-cultural interaction. Equivocation is the idea that ILK and science can find translations that preserve differences and non-equivalencies while producing similar *meanings* for both parties. An equivocation is a form of mistake, a "second-level" mistake. A "first-level" kind of mistake is to assume that there is a universal, correct, or superior set of perceptual categories—something that ethnobiology has disproven. An example is Dupré's study [43] showing how a particular folk taxonomy category of "fish" includes whales. "Correcting" fishermen to inform them that whales are not fish would miss that "fish" in this context is being used as a functional category akin to "omnivore" or "shrub". The ethic of equivocation operates at a "second level" in which two different forms of categorization or conceptualization are, in some context, treated by their respective users as equivalent, even if both parties know that each concept has other, non-equivalent aspects or ramifications. This is similar to the question of which appendage you shake when "shaking hands" with a lobster. The fact that one of its appendages will work perfectly well for a handshake does not mean that anyone actually believes that it is anatomically a hand.

Fitzgerald et al. [44] also describe equivocation as a working method for interdisciplinary research. They describe the dynamics of the collaboration between a group of social scientists and a group of neuroscientists at the same university. They were able to work together, exchange knowledge, and complete research, for the purposes of the project, without claiming to form a universal or stable set of equivalencies between their two knowledge systems. As this example illustrates, the ethic of equivocation is not limited to or predicated on researcher-informant relationships but works in any situation where there is dialogue, including interdisciplinary collaborations, applied actions, and co-generation of knowledge. The outcome of this interaction is multiple lines of evidence embedded in different epistemologies, which generate different interpretations according to peoples' worldviews. What makes equivocation unique is the focus on an interactive process that allows the coexistence of different understandings of key areas of overlap, without reducing, or subsuming, those differences into a unitary or global framework.

Achieving science and conservation goals through processes of dialogue, collaboration, or co-production does not require establishing points of direct and absolute equivalency,

or cross-validation. The ethic of equivocation allows space for the creative and generative potential of second-level mistakes that work, such as a genetic mutation that makes a flower look similar to a female bee and leads to better cross-pollination. The mutation obviously does not make the flower *turn into* a bee, but both the bee and the flower benefit from a male bee treating the flower as a female bee, and from the flower treating the mate-seeking bee as a potential pollinator.

## 4. Standard Methods and How These Might Be Done Differently in an Ethic of Equivocation

Standard methods for working with ILK may be divided into documentation, merging with science, and co-production of new knowledge [45]. Self-validating multiple evidence bases may be woven into any of these methods. Many studies see ILK as an information resource to be documented. They are thus concerned with issues of comparability or homogeneity of data standards, asking whether the information from ILK is precise and accurate and whether this can be validated scientifically (e.g., [46–49]). Following this reasoning, some researchers still feel they cannot consider ILK that have not been scientifically validated, or directly generated through a standards-based protocol [50]. In their research, scientists exclude their personal observations that they perceive as unvalidatable and may apply the same reasoning to ILK. Merging-with-science approaches typically also share the reasoning and approaches of the documentation approach, but go beyond documentation to use validated ILK to complement or constitute scientific datasets within projects designed (primarily) by scientists for scientific (e.g., statistical) analyses (e.g., [51,52]).

From an ethic of equivocation perspective, when considering data validation and dataset archiving, we see two issues for further consideration. Ecologists and conservationists might consider whether ILK has the same *meaning* to ILK holders when integrated into datasets and archives. Meaning is about *function*—not in the sense of a teleological or ultimate purpose, but in the ecological sense of the effects of a practice or outcomes of a process. Just as a detritivore has the function of decomposition, and plants have the function of fixing carbon, meanings have social functions. Within science, examples of meaning-making include the analysis of data, the framing and interpretation given in the Introduction and Discussion sections of papers, and the claims made about the importance of findings. The function of those meanings is to allow further research, understanding, and applications. So if we ask whether a piece of ILK-as-data; that is, a given dataset derived from ILK, has the same meaning in a given social context as that piece of ILK had (or continues to have in parallel) as not-data, we are asking whether ILK-as-data can *do* the same things e.g., to inform people's resource use practices or adaptive capacities, as in its original non-data form. For example, if a hypothetical society uses personal narratives to incorporate knowledge of weather into crop planting decisions, do quantitative data about past weather do the same things in that society? Are they used to make similar arguments? Do they lead to similar decisions? This leads to the second issue for further consideration, which is, of what use to conservation and ecology are aspects or sections of ILK that are *not* reformatted as data?

Aspects of ILK that are not converted into data that the ecologist or conservationist encounters for example being used by ILK-holders in the field, can be treated in several ways. In some cases, an established equivocation is available for a specific discrete outcome of ILK. For example, when an Indigenous tracker identifies an animal track whose location and ID are then saved into a GPS, the particular observation of the track is transformed into scientific data. In other cases, an equivocation can allow a scientist an opportunity to make novel scientific data directly, through overlaps in the *processes* of knowledge-making. For example, while a tracker and a wildlife biologist may have different explanations or uses for animal movements, the tracker's superior tracking skills help the wildlife biologist identify sites where certain samples can be obtained.

Finally, a third possibility is that ecologists and conservationists can treat pieces of ILK that are not data in the same way that fieldworkers treat other observations of study

phenomena not carried out in the framework of a scientific protocol. Things that don't immediately fit one's disciplinary knowledge, or that are akin to what scientists generally consider to be their unvalidatable personal observations, can be signs of interesting new research areas. They do not immediately yield data, but rather questions or hypotheses. The examples of Charles Darwin and Alfred Wallace (among others) remind us that detailed observation and comparison can lead to theory development [53]. The increasing number of natural history formats in scientific journals also reflects the growing recognition that exploratory research is a necessary complement to confirmatory research.

The ethic of equivocation approach leads us to argue that both documentation and merging approaches should constantly be on the look-out for opportunities to do the third approach, 'co-production of knowledge' [45,54], which can be defined as "a process that brings together diverse groups to iteratively create new knowledge and practices" [55] that are "actionable . . . credible, legitimate and salient to decision-makers" [56]. Work on sustainable development and community conservation are subfields that have embraced this approach (e.g., [55,57–60]). Co-production is often aimed at applied management contexts and strives to produce concrete benefits for all parties; it is, thus, about situated knowledge. Documentation-focused and merging-with-science work with ILK should remain attentive to the large mass of situated knowledge in forms other than data that may be more effective for certain kinds of problems, and which may inspire novel hypotheses, analytical categories, and research questions (see e.g., [61–65]). However, since all of the goals of ILK holders and scientists will not be common goals, even after the development of an epistemological relationship predicated on an ethic of equivocation, either party may also continue in parallel to produce knowledge for "internal consumption" within their discipline or tradition.

In summary, the ethic of equivocation approach insists that ILK is always richer, and often more motivating and more generative of novel hypotheses than is captured by its reduction to data. It invites the conservationist or ecologist to consider that all conversions to data are first-level mistakes with the potential to become second-level mistakes.

## 5. How to Do Equivocation in Practice

When conservationists and ecologists look "behind" or "beyond" the data they can validate from ILK, they may find things they do not know exactly what to do with. How can an ecologist or conservation biologist react when their informants tell them, for example, about taboos, spirits, magic spells, or myths? These are concepts, units, or materials that many scientists are not used to including in their research.

Some aspects of ILK seem impossible to validate—what a scientist might identify as a belief rather than a fact supported by evidence. Of course, facts should also be doubted: in science, theories and facts are never proven, only disproven. However, scientists also think with and about ideas that are not facts (nor beliefs), such as classic predator-prey models that do not accurately predict dynamics, the intermediate disturbance hypothesis, or metaphors, such as the selfish gene or the tree of life [6]. The scientific issue is whether such models, concepts, and metaphors are useful for thinking about a particular problem or topic [66]. Faced with (what appear to be) others' models, concepts, metaphors, or beliefs lacking (convincing) evidence, one can usefully ask, what do they mean? To answer such questions, many anthropologists think about the meanings of beliefs/facts/knowledge by attending to their *effects on how people act in the world*.

An ethic of equivocation approach to thinking about *how ILK affect how people act in the world* is to find a scientific approach, format, or formalism, which allows (in practice) or represents (in analysis) *a similar effect*, and to use that as the basis for dialogue and knowledge co-production.

Concretely, an equivocation can be a concept, a formality, a practice, or an institution. Furlan et al. [41] provide several examples of equivocations, mainly around concepts and practices that arose during their ethnobiological research in South America. Here, we highlight several other examples. For example, in the Andean Kichwa culture, there is a concept

called *Sumak Kawsay*. While this has been widely translated and popularized with the Spanish term *buen vivir* ('living well') [67], the original concept is much broader, with a richer set of implications, than suggested by this translation. Although conservationists arguably misunderstand this when they equate *buen vivir* to "sustainable development" [68], this misunderstanding may be a productive starting point for working towards common goals.

Congretel and Pinton [69] provide an example of equivocation as a formality. Although the meaning(s) of guaraná (*Paullinia cupana*) is/are radically different between Indigenous cultivators (as a sacred food source) and industrial producers for export (as a coffee-like stimulant), the co-development of a formal document describing how to produce guaraná gave legitimacy to both understandings while allowing conservation cooperation. The formality has not reduced the number of understandings of guaraná circulating in the world, but it has allowed different parties to work together towards the common goals that it enshrines and facilitates. Legal rulings about the personhood of geographical entities or biocultural rights, and other protocols and agreements protecting communities' knowledge and rights provide similar examples [70].

At the intersection of practices and formalities, one well-described example of equivocation is how early colonial settlers in New England understood property rights, and how they interpreted whether local Indigenous groups had property rights [71]. Colonists were often explicitly aware that their ideas about rights, usufruct, and ownership of land did not correspond to local Indigenous concepts and practices. Although some arrangements regarding sale/land sharing may initially have seemed satisfactory to both parties, this equivocation about land rights and usufructs over time came to be deliberately misrepresented by the colonists to their political advantage. These deliberate misrepresentations, which can be contrasted with honest equivocations, had direct ecological impacts due to land-use change on lands taken over by colonists, as well as due to Indigenous groups' being forced to find alternative land-use practices in reduced spaces impacted by surrounding land-use changes. Ultimately, satisfactory and productive equivocations were abandoned as colonists promulgated deliberate misunderstandings to justify the mass displacement and genocide of Indigenous peoples. This example thus also highlights the difference between an equivocation, which all parties accept, and a deliberate misunderstanding which is dishonest, exploitative, and not mutually accepted.

Finally, an entire institution can be the focus of an equivocation. Protected areas, for example, may be institutional arrangements of which different stakeholders have widely divergent perceptions and expectations, e.g., they should prevent industrial projects, bring economic revenue, or increase certain species' populations. Some protected areas achieve outcomes that are satisfactory equivocations for multiple parties, others do not (e.g., [72,73]).

## 6. Learning How to Do Equivocation Well

The reader will note from the preceding section that equivocations require honest engagement from both parties. A mutually honest and respectful equivocation is not immune to being abandoned and replaced by a deliberate and exploitative misrepresentation. This is why the ethic of equivocation may be referred to as "controlled equivocation", since if not done attentively it may also escape control and be exploited by malicious actors [38,41]. The importance of attention and care is also why it is an "ethic" and not just a method or procedure. Although we cannot force people to be or remain ethical, we argue that without practicing the ethic of equivocation scientists may (unintentionally) incur ethical problems in their relations to ILK holders. As we explain above, asserting that a piece of scientific knowledge and a part ofILK are exactly equivalent or radically incomparable are both untenable political positions with suboptimal outcomes for knowledge creation. Accepting that we will make mistakes, and trying to make good, productive mistakes via the ethic of equivocation, is, we suggest, the best course of action. As we discuss below (Section 7: **Issues of power relations in epistemological relationships**) the worst outcomes can be avoided through respect, attention, taking seriously the claims

of one's collaborative partner, and through appropriate political representation within decision-making contexts. Experience over time, training in transdisciplinarity, and personal engagement with the culture of ILK holders may also help [74]. It can also help to learn from successful, comparable situations.

Collaborations are a perhaps more-familiar place for ecologists and conservation biologists to start looking for situations from which to learn. Collaborations in general, and interdisciplinary collaborations in particular, pose some similar issues to the ethic of equivocation in an ILK context, and advice and reflections on researchers' struggles to collaborate may provide some inspiration (e.g., [44]). However, not only is collaboration generally messy and less than ideal, but working across (sub)disciplines even within science/academia cannot be reduced to a list of best-practice steps to follow.

Another (and even more pertinent) source of inspiration for working with ILK and ILK-holders comes from ILK-holders working with and/or as scientists. Many cross-cultural complexities have been described from the perspective of ILK-holders engaging science to develop Indigenous research methodologies (e.g., [75–77]).

ILK-holders are experts in the merging of science with ILK and may provide concrete examples that we interpret as equivocation in practice. For example, Berenice Sanchez reports, "Indigenous Peoples have always depended on the full understanding of the seasons to survive and have accumulated knowledge through the centuries based on observations. Everything, absolutely all the factors, has been observed and interpreted . . . What we offer is science because we have systematized and interpreted these observations" [78] (p. 22). Eddie Haikau Huitarau writes, "Although the Ahetaha community [of the Soloman Islands] has been going through lots of changes in adopting a more Western lifestyle, our traditions remain part of us, like our traditions of governance, shifting cultivation, bridal dowry, mourning rituals, and the conservation method known in our dialect as *Asiia*. . . . We encourage our children to attain education and become specialists in areas of conservation already in place." [79] (pp. 20–21). Hindou Oumarou Ibrahim has stated, "Where our traditional knowledge and science meet, we say that climate science uses modern knowledge . . . But for us, we are just using our information and other factors we observe . . . When we break [a fruit], we see the liquid inside. If the liquid is abundant, we know [what] the tree is predicting for the next year." [80] (p. 11). These ILK holders have already integrated scientific concepts into their knowledge systems. They are *holding multiple representations in mind simultaneously*, and in some cases claiming these as points of what we describe as equivocation, e.g., claiming that what they are doing is science; or that they become 'Western' scientists *in order to keep doing what they were already doing*. Within an ethic of equivocation, they may simultaneously self-validate their knowledge (from their perspective), and validate their knowledge from a science perspective; but no cross-validation occurs.

## 7. Issues of Power Relations in Epistemological Relationships

There are many relevant issues around the balance of power that deserve attention when developing an epistemological relationship [30]. The historical case of colonists' interpretations of land rights in New England also illustrates how a political power imbalance, in which both parties are not mutually considered and treated as equals, can allow an initially valid equivocation to be replaced by a justification for exploitation and even genocide [71]. However, equal political representation by itself does not automatically translate into satisfactory outcomes. As we discussed above, the idea that the synthesis of ILK and science constructs common knowledge through a neutral synthesis process (in which no one's ontology or cosmology is implicated) is a false and unrealistic pretense. However, as in the example of the formal document governing *guaraná* production [69], it can be possible, through a sustained process, to develop joint understandings that can be simultaneously *interpreted in different ways* and *represented in different forms* within multiple ontologies and cosmologies in ways that lead to coordinated actions that are satisfactory to all parties.

The development of satisfactory equivocations takes time, and scientists should not imagine that they will succeed rapidly and immediately. Developing new equivocations for novel purposes is unlikely to work within short-term engagements with a field site. Indeed, the search for "efficient" processes for arriving at satisfactory equivocations may lead to poor outcomes and abuses of power. Abuses of power in the development of such an epistemological relationship can consist of, for example:

- Insisting that two things *are* the same and refusing to acknowledge that any claim of equivalence must be to some degree a mistake. The acknowledgment of a latent degree of mistakenness allows the process to continue to progress towards more satisfactory, or differently satisfactory, equivocations. The insistence that two things are the same can be interpreted as an attitude, whether intended or not, of reductionistic closed-mindedness that may impede trust, cooperation, and the development of new knowledge.
- Insisting that two things are satisfactory equivocations when the other party does not agree; imposing the conditions for agreement about satisfactory equivocations on the other party.
- Taking a satisfactory equivocation and using this as a justification to impose a scientific worldview on related subjects because "a point of equivocation has been established"—in fact, each subject requires its own equivocation. You might think of this as similar to the problem of triangulating two satellite images in GIS in order to overlay them—but since ontologies are much more complex than two-dimensional planes, you need many more than three points to make this alignment. Certainly, it will take time to extend the field of satisfactory equivocations to a wide range of subjects. Acknowledging that it remains unestablished how other ILK and science concepts, practices, formalisms, or institutions relate to one another during this process is a good starting point.
- Taking an initial equivocation and failing to engage in some form of ongoing dialogue, collaboration, or other mechanism bringing parties together and allowing correction and adjustment of the equivocation over time and across contexts. Without such a process, the possibilities increase over time for obfuscation, mystification, dishonesty, abuse, and so on.

There are also numerous issues related to methods, acknowledgment and credit, authorial representation, benefit sharing, and many other issues of politics and collaborative ethics, which have been treated by other authors in great depth (e.g., [4,81–85]), and by research guidelines issued by Indigenous communities themselves. Many of these issues need to be dealt with in ways specific to particular communities.

Our general advice is that if, as a scientist, you want to work with ILK but you have the feeling that the local community does not want you to be there doing that, you should stop doing it. One should not fool oneself that any ecological question or conservation problem absolutely requires one's personal scientifically-informed intervention to solve it. Some communities do not want outside ecologists and conservation biologists to collaborate with them; others welcome them. Our second piece of general advice is that scientists choosing field sites and working methods should do background research on the societies where they want to work, in order to determine whether there are current tensions around post-colonialism, neo-colonialism, or other political issues that might put a scientist who either engages (in specific ways) or does not engage (in specific ways) with ILK holders in a politically tense or complex situation [85]. Individuals who feel strongly that they have adequate social and intercultural skills to deal with tense political situations may thrive and do good work in such contexts; other scientists who judge that they lack these skills may find it more productive to work with "local ecological knowledge" in their societies of origin, or in areas where such tensions are more resolved or less developed. Even there, ethical concerns about one's position as a scientist relative to society do not disappear. As a third and final general piece of advice, the ethic of equivocation is an approach to epistemological relationships that embraces ambiguity and uncertainty. No one should

feel obliged to develop epistemological relationships that make them, or the people they want to interact with, uncomfortable. In this case, however, if one chooses to avoid the ethic of equivocation in one's own work, one should be doubly attentive to not fall into the epistemological abuses of power described above.

## 8. Conclusions

Although collaborative approaches combining ecologists, conservationists, anthropologists, and ILK holders are very valuable, here, we demonstrate to ecologists and conservationists that they may themselves have more tools than they are aware of with which to directly engage in sophisticated ways with ILK holders. Throughout this paper, we point to these intellectual tools by using several analogies and metaphors that may help to clarify what ILK are and how the ethic of equivocation suggests methods for engaging with ILK. Pluralism in styles of reasoning and working already exists in science (e.g., [86]), and a first step to engaging with ILK is to become self-aware of one's own style, and to acknowledge and potentially mobilize different styles. Most reasonable scientists recognize that styles of argument and method, subdisciplines, or disciplines cannot be simply reduced to correct or incorrect, but have points of disagreement, contact, comparison, divergence, and convergence. Engaging with ILK can be seen as a profound and radically challenging extension of interdisciplinarity or transdisciplinarity, given the differences in underlying worldviews that provide basic interpretive schema. There is no perfect correspondence between any ILK and any academic discipline, no Rosetta Stone, and no perfect way to engage across them, which is why an ethic of equivocation based on second-order mistakes forming contextually functional equivalencies can be a respectful and productive way forward.

We conclude with a final analogy. We suggest that a productive way to imagine a science-ILK collaboration is along the lines of a collaboration between two artists from different genres. As an example, consider the performance of "The Dying Swan" by Lil Buck and Yo-Yo Ma (https://www.youtube.com/watch?v=C9jghLeYufQ, accessed 23 October 2020). In this example, neither Lil Buck nor Yo-Yo Ma is a perfect analogy for either a scientist or an ILK holder. Lil Buck is a dancer in the Jookin style, a popular dance from Memphis, USA. Yo-Yo Ma is a classical cellist, part of an elite European tradition. The rigorously trained cellist's prestige within the classical music world was not lessened, but rather increased, by his association with the self-taught and, at the time, relatively unknown dancer, and vice versa. The performance brought new dimensions to the work of each artist, creating a novel whole that was startling and exciting to millions of observers. An ethic of equivocation across ILK and science can result in equally pathbreaking advances in knowledge of and care for our natural world.

**Author Contributions:** M.R.-B. conceived the presented idea. All authors contributed equally to the writing and editing of the paper. All authors have read and agreed to the published version of the manuscript.

**Funding:** This research received no external funding.

**Institutional Review Board Statement:** Not applicable.

**Informed Consent Statement:** Not applicable.

**Data Availability Statement:** No data were produced or used in this paper.

**Acknowledgments:** M.R.-B. considers this as a contribution to the Jefferson Fund grant no. TJF18_067. S.R. and M.R.-B. consider this as a contribution to the ECOS-Sud grant C18B0.

**Conflicts of Interest:** The authors declare no conflict of interest.

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
