# Peer review of "What Are ILK in Relation to Science? Using the ‘Ethic of Equivocation’ to Co-Produce New Knowledge for Conservation"

_sustainability, doi:10.3390/su15031831_

Round 1
Reviewer 1 Report
The topic of the manuscript is fresh and refreshing. Although at times monotonous, a lot of new information is being provided that is ably supported by citing literature. I suggest acceptance of the article in its current form.
Author Response
Thank you for recommending acceptance, we appreciate this positive feedback.
Reviewer 2 Report
The MS touches upon a critical question about the relationship between ILK and traditional science. It is very well written and quite logical. I don't have any major concerns. The only minor comment is that the conclusion section seems to lack discussions around the suggestions for engaging with ILK. I believe this could potentially be an important take-home message that will inform readers of how to treat ILK rigorously. I recommend the authors give more highlights on their recommendations in the revision.
Author Response
We have added a summarising paragraph to the Conclusions to address the suggestion to add "discussions around the suggestions for engaging with ILK." Thank you for the positive view of the paper.
Reviewer 3 Report
See the attachment for some detailed corrections.
In some areas, the text is written with different fonts.
A challenging paper! Congratulations!

Author Response
We have fixed the spelling and typing errors and correctly formatted the paper. Thank you for the positive appreciation of the paper.
Reviewer 4 Report
paper is nicely written except some mistakes which can be improved
Content of the paper is the need of the hour
Author Response
We have fixed some spelling and typing errors and correctly formatted the paper. Thank you for the positive remarks.